# Reviewing Atrial Fibrillation Pathophysiology from a Network Medicine Perspective: The Relevance of Structural Remodeling, Inflammation, and the Immune System

**DOI:** 10.3390/life13061364

**Published:** 2023-06-10

**Authors:** Ivis Levy Fernandes Martins, Flávia Valéria dos Santos Almeida, Karyne Pollo de Souza, Fernanda Carla Ferreira de Brito, Gabriel Dias Rodrigues, Christianne Bretas Vieira Scaramello

**Affiliations:** 1Research Nucleus on Plasticity, Epidemiology and In-Silico Studies (NUPPEESI), Fluminense Federal University, Niteroi 24020-141, Rio de Janeiro, Brazil; ivisfernandes@id.uff.br (I.L.F.M.); flaviavaleria@id.uff.br (F.V.d.S.A.); karynepollo@id.uff.br (K.P.d.S.); 2Experimental Pharmacology Lab (LAFE), Fluminense Federal University, Niteroi 24020-141, Rio de Janeiro, Brazil; brito_fernanda@id.uff.br; 3Experimental and Applied Physiology Lab (LAFEA), Fluminense Federal University, Niteroi 24020-141, Rio de Janeiro, Brazil; gabrieldias@id.uff.br; 4Department of Clinical Sciences and Community Health, University of Milan, 20126 Milan, Milan, Italy

**Keywords:** atrial fibrillation, bioinformatics analysis, systems biology, network medicine, immune cell, inflammation, structural remodeling

## Abstract

Atrial fibrillation (AF) is the most common type of sustained arrhythmia. The numerous gaps concerning the knowledge of its mechanism make improving clinical management difficult. As omics technologies allow more comprehensive insight into biology and disease at a molecular level, bioinformatics encompasses valuable tools for studying systems biology, as well as combining and modeling multi-omics data and networks. Network medicine is a subarea of network biology where disease traits are considered perturbations within the interactome. With this approach, potential disease drivers can be revealed, and the effect of drugs, novel or repurposed, used alone or in combination, may be studied. Thus, this work aims to review AF pathology from a network medicine perspective, helping researchers to comprehend the disease more deeply. Essential concepts involved in network medicine are highlighted, and specific research applying network medicine to study AF is discussed. Additionally, data integration through literature mining and bioinformatics tools, with network building, is exemplified. Together, all of the data show the substantial role of structural remodeling, the immune system, and inflammation in this disease etiology. Despite this, there are still gaps to be filled about AF.

## 1. Introduction

Atrial fibrillation (AF) is the most common type of sustained arrhythmia, which encompasses irregular high-frequency excitation and contraction of the atria, and affects 1–2% of the general population, particularly the older population. AF is silent in 5–35% of diagnosed patients, so its overall prevalence may be higher [1,2].

AF can be paroxysmal, occurring in about 50% of cases, or persistent, present in 20% of chronic heart failure cases, with a poor prognosis [3,4]. AF is related to a significant financial burden to patients and the healthcare system, as patients with AF are 30% more likely to be hospitalized at least once per year [5]. This condition is also associated with reduced quality of life and a high risk for dementia, embolic stroke, and elevated mortality, contributing to an increased chance of sudden death. AF raises both men’s and women’s mortality, with a 1.5- and 1.9-fold increase in the odds ratio, respectively [6,7,8].

Although the underlying mechanisms of AF remain unclear, studies have shown that the pathogenesis is multiplex, assuming genetic and environmental factors. It includes electrical and structural remodeling, Ca^2+^ handling, and autonomic nervous system abnormalities. Thus, this condition may be a final common endpoint of atrial remodeling or electrical, structural, cardiac autonomic nerves, and ion channel changes that cause irregular atrial rhythms and unique spatial heterogeneous conduction [9,10]. An improved understanding of AF mechanisms would enable the development of better clinical management.

To date, a standard therapy embraces ablation, including pulmonary vein isolation with or without atrial substrate ablation. This invasive procedure presents limited success, with relevant recurrence rates and the risk of significant complications [11]. With sinus rhythm stabilization, b-adrenergic blockers, calcium channel blockers, and cardiac glycosides can control heart function. However, it is essential to highlight that the side effects of antiarrhythmic drugs limit rhythm control, such as life-threatening arrhythmia. Anticoagulants with or without left atrial appendage closure prevent thromboembolism, carrying bleeding, and procedure risks [12,13]. According to the network meta-analysis developed by Wang, Tingting Fang, and Zeyi Cheng (2022) [8], non-pharmaceutical therapies were superior to traditional pharmacotherapies in efficacy, reducing arrhythmia recurrence/re-hospitalization, and safety, diminishing ischemic cerebral vascular events, all-cause mortality, and cardiovascular mortality.

Despite this, it is widely acknowledged that the clinical management of AF requires improvement. There are advances concerning the knowledge of its mechanism, but also numerous gaps [2,14,15,16]. Generally, a single or limited number of molecular mediators are linked to a disease phenotype. The approach ‘one disease–one target–one drug’ is reductionist, giving a limited view/results on complex human conditions [17].

Omics sciences involve genomics, transcriptomics, proteomics, and metabolomics. Omics technologies allow a more comprehensive insight into biology and disease at a molecular level [18]. The complete set of molecular interactions within a cell (gene–gene, gene–protein, protein–protein, and others) is known as the interactome, and characterizes systems biology [19,20].

It is possible to create a wide range of biological systems through networks. Network biology involves biomedical applications of network theory. Nodes represent distinct individual biological entities, and the relationships among them, known as edges, are depicted by connecting lines. These interactions may encompass gene regulation, physical protein–protein interaction, or substrate metabolism. Bioinformatics tools are available to combine and model multi-omics data and networks. This approach interprets disease traits as perturbations within the interactome, such as genes, RNAs, or proteins differentially expressed between patients and controls. Despite possible limitations due to the sample size, microarray analysis, or incomplete interactome, network medicine, a subfield of network biology, largely contributes to the study of disease etiologies from an omics and analytical point of view, enabling biomarkers and drug discovery [17,21,22].

This work aims to review AF pathology from a network medicine perspective, helping researchers to gain a broader comprehension of the disease, its diagnosis, and promising treatment. Thus, essential concepts involved in network medicine will be highlighted, and specific research applying network medicine to study AF will be discussed. Finally, data integration through literature mining and bioinformatics tools, with network building, will be exemplified.

### 1.1. Disease Etiology Studies through Molecular Networks

Network biology integrates systems biology, graph theory, and computational/statistical analyses. Thus, integrated omics data are represented by nodes and edges. It is possible to analyze the resultant network topologically. Various algorithms and parameters are also involved in modeling and predictive or differential analysis [17,23].

Topological parameters describe the metrics and quantitative patterns of edges and nodes. Centrality parameters help reveal relevant biological entities. For example, highly connected nodes are called hubs, and play critical roles in biological processes. The number of interaction partners constitutes the degree of centrality, being a local centrality parameter. The edges linking the nodes form a path; the minimum number of edges needed to connect two nodes encompass the shortest path length parameter. The betweenness centrality expresses how often a given node acts as a link on the shortest paths among all pairs of possible nodes. This parameter may identify low-degree nodes whose removal is fatal to the organism. Closeness centrality is another local centrality parameter that comprehends flow routers. It represents the average shortest path from one node to all other nodes, representing the speed of information conduction from a particular node to further accessible nodes. According to the literature, the disruption of central nodes can be linked to cancer, suggesting that they act as information coordinators. Network medicine is a subarea of network analysis that studies disease traits in the interactome and their possible treatment [17,22]. The term ‘network pharmacology’ can be alternatively applied when the study focuses on pharmacological treatment [20].

### 1.2. Network Medicine and AF

In network medicine, disease modules embrace localized perturbations in the network that characterize diseases. Thus, biological entities mechanistically define pathological signaling by their neighborhood in the interactome (disease signatures). With this approach, potential disease drivers are revealed, and the effect of drugs, novel or repurposed, used alone or in combination, may be studied, allowing advances in the knowledge of disease mechanisms and precision therapeutics [17,20,23,24,25].

Figure 1 summarizes the perspective of network medicine on the health–disease process. This knowledge gives excellent opportunities for rational drug development and repurposing.

Thus, this review aims to present network medicine as a novel approach to help researchers understand AF more deeply, describing the main related findings in the literature and demonstrating how bioinformatic tools can provide a holistic perspective of the disease’s underlying mechanisms, favoring treatment improvement.

## 2. Materials and Methods

To attain the primary goal of the present article, we first performed a narrative review of the published papers on AF studies from a network medicine perspective. The search was carried out using the Medline (PubMed) database. The terms used were “Atrial Fibrillation” or “Atrial Fibrillations” or “Auricular Fibrillation” or “Auricular Fibrillations” and “network analysis” or “biology systems” or “network medicine”. This search yielded 302 classical articles in English published between 1 January 2017 and 17 February 2023. Analysis of the title and the abstract of the papers composed the preliminary screening. In the end, about 28 articles were included.

After this, aiming to briefly demonstrate how bioinformatic tools can provide a holistic perspective of the disease’s underlying mechanisms, data from these papers were integrated using a tool for the automatic annotation of files and biological information extraction named OnTheFly version 2.0 [26,27]. After selecting the top 30 biological entities cited in at least five articles, the related information about them was obtained by consulting the Uniprot database [28]. Enrichment analysis and network construction were also performed using OnTheFly version 2.0. Regarding the enrichment analysis of the top 30 biological entities, the Gene Ontology (GO) database in OnTheFly version 2.0 gives information about gene products from all organisms. A biological process involves a particular set of molecular processes conducted by specific gene products. GO terms typically describe functions of gene datasets subjected to statistical tests for enrichment for various functional categories. Gene datasets can indicate that a systems-level change in a particular condition has occurred, which may be described by examining the common properties of the differentially expressed genes. The GO enrichment analysis may answer whether the genes participate in the same metabolic or signaling pathway and perform similar biochemical functions. Here, we present enrichment analysis related to ontology pathways [29].

## 3. Diseasome and Non-Coding RNAs in AF Pathogenesis

In the context of network medicine, it is also necessary to highlight the concept of diseasome, where nodes represent diseases linked to each other by sharing a common genetic component. Furthermore, diseases may be clustered based on shared risk genes. For instance, as the prevalence of AF is higher in patients with lung cancer, Yan et al. (2022) studied the association between these two conditions [30]. After identifying AF-related genes, the authors analyzed the expression profiles, prognosis, immune infiltration, and methylation characteristics of these genes in patients with lung adenocarcinoma, the common histological subtype of lung cancer. They found several AF-related genes, including CBX3, BUB1, DSC2, P4HA1, and CYP4Z1, differentially expressed between tumor and normal tissues. While CYP4Z1 was positively correlated with overall survival in these patients, CBX3, BUB1, DSC2, and P4HA1 were negatively correlated. Additionally, a hub immune-related gene in AF related to cancer named ANXA4 had been previously identified. Thus, ANXA4 might play crucial roles in AF and cancer, and targeted therapy for ANXA4 might reduce the incidence of AF in cancer patients [31].

Among all biological entities forming the interactome, the probable involvement of non-coding RNAs in AF pathogenesis is becoming apparent. MicroRNAs (miRNA) are small non-coding RNAs regulating gene expression at the post-translational level. There are miRNA binding sites on messenger RNAs (mRNAs), long non-coding RNAs (lncRNAs), circular RNAs, and pseudogenes, suggesting that RNA transcripts containing miRNA-binding sites may modulate each other by competing for shared miRNAs, playing the role of competing endogenous RNAs (ceRNAs). The existence of ceRNA crosstalk in AF pathogenesis is discussed. The literature shows that several miRNAs are involved with atrial electrical and structural remodeling. The protagonism of non-coding RNAs in disease pathogenesis, especially lncRNAs, is being widely investigated. For example, FAM201A (the family with sequence similarity 201 member A) is an RNA gene affiliated with the lncRNA class. The decrease in lncRNA FAM201A expression is related to mRNA RAC3 downregulation favoring autophagy, which could lead to a decrease in the L-type calcium channel expression and current, shortening the action potential duration [32]. Likewise, cardiac apoptosis-related lncRNA competitively binds to miR-539, preventing the miR-539-dependent downregulation of the protein PHB2. Consequently, PHB2 can inhibit mitochondrial fission and apoptosis in cardiomyocytes [33]. Many network medicine studies show the relevance of irreversible structural remodeling, besides electrical remodeling, for the occurrence, development, diagnosis, and treatment of AF.

Liu et al. (2021) screened the top ten hub nodes with the highest degrees related to AF genesis and progression by analyzing a protein–protein interaction network built from gene expression profiles of the GSE41177 dataset. They identified autophagy-related hub genes and revealed six significant immune cell subpopulations. Six differentially expressed autophagy-related genes (BECN1, GAPDH, ATG7, MAPK3, BCL2L1, and MYC) and three immune cell subpopulations (T cells CD4 memory resting, T cells follicular helper, and neutrophils) were identified as the most significant potential regulators [34]. Yang et al. (2021) and Qu et al. (2021) also used WGCNA to identify hub mRNAs and non-coding RNAs in AF [35,36]. Comparing samples from subjects with and without AF, differentially expressed genes (DEGs) regulate multiple pathways, such as the modulation of vascular endothelial growth factor production and binding to the CXCR chemokine receptor. Among all DEGs, about half were non-coding RNAs. Five hub modules were significantly negatively and positively correlated to the progression of AF. Besides the hub lncRNA–mRNA regulatory network, the hub protein–protein interaction (PPI) network was constructed and analyzed, showing the relevance of immune response signaling, such as leukocyte chemotaxis, macrophage activation, and positive regulation of α-β T cell activation in AF [35]. Qu et al. (2021) identified two significant modules and two hub miRNAs (hsa-miR-146b-5p and hsa-miR-378a-5p) highly correlated with the AF phenotype and differentially expressed miRNAs and genes were predominantly enriched in inflammation-related functional items. By overlapping the DEGs and predicted target genes, nine genes (ATP13A3, BMP2, CXCL1, GABPA, LIF, MAP3K8, NPY1R, S100A12, and SLC16A2) located at the core region in the miRNA-gene interaction network were identified as hub genes [36]. Thus, structural remodeling, inflammation, and the immune system seem relevant in AF pathogenesis.

## 4. Structural Remodeling

Zou et al. (2018) identified DEGs and hub genes, including LEP, FOS, EDN1, NMU, CALB2, TAC1, and PPBP, in AF. The analysis revealed that the maps of extracellular matrix (ECM)–receptor interactions, PI3K-Akt and Wnt signaling pathways, and ventricular cardiac muscle tissue morphogenesis were significantly enriched. They concluded that the ECM–receptor interaction is probably a central node closely associated with AF duration. Furthermore, the occurrence and maintenance of the disease may implicate the hub genes [37].

A meta-analysis of transcriptomic data revealed that 1197 DEGs are involved in the pathophysiology of AF. Hub bottlenecks and modules enriched in several biological pathways were identified, especially those related to the disease’s structural and electrical remodeling processes [38]. This meta-analysis included the work of Liu et al. (2020) [39]. These authors have found that critical genes such as SPP1, COL5A1, and VCAN are upregulated in AF tissues, concluding that genes related to the ECM are involved in the pathology of AF being promising targets for the diagnosis and treatment of this disease. Yu et al. (2021) also investigated the relationship between ECM organization and AF progression, besides metabolic dysregulation and the vitamin D response. They observed that hub genes, such as APP, CDH2, SPP1, and STC2, are differentially expressed in this condition [40]. Many other studies also point to structural remodeling in AF pathophysiology due to interactome perturbations.

Recently, more AF-associated differentially expressed genes were mapped. Downregulation of ERBB2 was related to the increase in oxidative stress and the worsening of mammalian heart regeneration and cardiomyocyte differentiation/proliferation. At the same time, the upregulation of MYPN and abnormal atrial myocardial fibrosis were linked. Numerous mutation loci in the gene MYPN are associated with hypertrophic, dilated, and restrictive cardiomyopathy [41]. Yu et al. (2021) also found DEGs related to fibrosis. They highlighted the CXCR4, TLR4, and CXCR2 genes, the Hippo signaling pathway, and the class A/1 (rhodopsin-like receptors) as having critical roles in AF occurrence and maintenance [42].

The study of Zou et al. (2022) also suggested mechanisms involved in fibrosis leading to AF, including the renin–angiotensin–aldosterone system, transforming growth factor-β1 (TGF-β1), oxidative stress and inflammation, calcium overload, matrix metalloproteinases, and microRNA. Matrix metalloproteinases (MMP) encompass a family of zinc-dependent proteolytic enzymes that affect extracellular matrices, including gelatinase, collagenase, and matrix enzymes. They identified six hub genes involved in the occurrence and maintenance of AF, including ST8SIA5, ODC1, LAPTM5, NPC2, SNAP29, and FCGR3B. Most of them are genes related to lysosomal autophagy. In addition, regulatory factors, such as EIF5A2, HIF1A, ZIC2, ELF1, and STAT2, were also significantly associated with the occurrence and maintenance of the disease through targeting candidate genes [43]. Likewise, Zhang et al. (2020) previously discussed the role of the TGF-β signaling pathway in AF. Comparing samples from AF and healthy individuals, they found twenty upregulated and three downregulated circRNAs, such as hsa_circ_0000075 and hsa_circ_0082096. In addition, the ceRNA network analysis revealed two upregulated genes, IFNG and GDF7, and one downregulated gene, BMP7 [44]. Thus, the TGF-β signaling pathway may be relevant to AF fibrosis.

Liu et al. (2021) screened crucial lncRNAs, miRNAs, and mRNAs for AF using different microarray datasets. About 120 module-related differentially expressed mRNAs and their hub genes were mapped. The researchers identified two novel lncRNAs (MIR100HG and LINC01105) associated with AF that may function by co-expressing with (MIR100HG-ROCK1/FBXW7/UBE2D1, LINC01105-EGFR) mRNAs or sponging miRNAs for mRNAs (LINC01105-miR-125a-3p-EGFR, MIR100HG-miR-200b-3p-FBXW7, MIR100HG-miR-561-3p-CXCR2) to regulate cardiomyocyte apoptosis and atrial fibroblast proliferation, ultimately leading to the development of the disease [45]. Using a similar approach, Ke et al. (2022) found eight lncRNAs and 43 mRNAs significantly differentially expressed in samples of AF patients compared to subjects with sinus rhythm. Enrichment analysis showed that cardiac muscle contraction pathways were involved in developing the disease. The expression of miR-490-3p was downregulated, while LOC101928304 and LRRC were upregulated in the myocardial tissue of patients with AF [46].

Liu et al. (2021) observed abnormal expression and/or methylation of the RHOA, CCR2, CASP8, PHLDA1, MUC4, PCDHA family, and SYNPO2L genes in AF through a transcriptomics study and WGCNA. Furthermore, the identified DEGs were enriched in the regulation of the actin cytoskeleton and in neutrophil activation [47]. Thus, besides cell morphology, the immune system may also be involved in AF.

## 5. Inflammation and Immune System

Several recent studies embracing network medicine have highlighted the relevance of inflammation in AF. For instance, Li et al. (2020) identified 20 modules performing WGCNA on an AF dataset. The most critical two presented six hub genes, ACAT1, CRADD, GIN1, FTX, TCEAL2, and MCM3AP, discussed as having critical roles in the pathophysiological mechanisms of the disease. One module was associated with energy metabolism, and the other with multiple complex interactive apoptosis and inflammation pathways [48]. Liu et al. (2021) identified the persistence-disease-associated module involved in the electrical remodeling and the susceptibility-disease module related to the inflammatory process. This study obtained RNA sequencing data from 235 left atrial appendage samples from the GEO database. The top lncRNAs/mRNAs with the highest variance were used to build a gene co-expression network by WGCNA, whereas MIAT and LINC00964 were identified as key lncRNAs [33].

Yan et al. (2021)’s study screened 2350 DEGs clustered into eleven modules using WGCNA. A key module with 246 genes was associated with M1 macrophages with the highest correlation coefficient. Three hub genes, CTSS, CSF2RB (common beta chain of the high-affinity receptor for interleukin-3, interleukin-5, and colony-stimulating factor), and NCF2, were found to be upregulated in patients with AF [49]. Ying et al. (2023) showed that regulatory T cells, resting natural killer cells, active mast cells, and neutrophils may be closely related to nonparoxysmal AF. The permanent subtype was associated with six hub genes, such as TYROBP, PTPRC, ITGB2, SPI1, PLEK, and CSF1R, while the persistent subtype was associated with five hub genes, such as JAM3, S100P, ARPC5, TRIM34, and GREB1L [50]. Thus, increasing evidence also demonstrates the significant role of the immune system in this condition.

In this scenario, network medicine researchers are developing several omics studies using bioinformatics to find inflammatory diagnostic markers and potential drug targets in AF, including those encompassing immune infiltration analysis, usually using CIBERSORT. There were higher levels of monocytes, dendritic cells, and neutrophils, as well as lower levels of CD8+ T cells and regulatory T cells (Tregs) in AF patients [51]. In addition, they identified several novel disease-associated genes, miRNAs, and pathways, including microRNA-34a-5p, which might target RCAN1 and PPP3R1 to regulate persistent AF through the calcineurin-NFAT signaling pathway. Likewise, Wei et al. (2022) showed that plasma cells and M2 macrophages significantly increase in AF, while follicular helper T cells and activated dendritic cells significantly decrease. Among 20 mainly related to muscle contraction, the autophagosome, and bone morphogenetic protein binding, and focused on the calcium signaling pathway, ferroptosis, and ECM–receptor interaction, the authors identified five potential key diagnostic genes. They were GPR22, COG5, GALNT16, OTOGL, and MCOLN3. Potential circRNA such as has_circ_0006314 and hsa_circ_0055387 were also shown to be biomarkers for postoperative AF, and immune genes such as CHGB, HLA-DRA, LYZ, IGKV1-17, and TYROBP were upregulated [52]. In addition, macrophages, mast cells, and neutrophils were significantly infiltrated in left atrial tissue [53]. Zheng et al. (2022) also observed more M2 macrophages infiltrated in AF patients and more gamma delta T cells and resting mast cells. However, there were fewer activated mast cells and regulatory T cells. Three novel hub genes were identified in this work. While BEX2 and GALNT16 were downregulated, higher levels of HTR2B were registered [54]. All these data together strongly suggest that immune cells may interact with specific genes in AF.

It is essential to highlight that both Chen et al. (2022) and Zheng et al. (2022) have also observed relevant modules enriched in inflammatory pathways in AF [53,54]. These data corroborate the work of Fan et al. (2020), who discovered 150 upregulated and 110 downregulated genes comparing samples from AF and sinus rhythm groups. Hub nodes such as CXCR4, CXCR2, C3, CXCL11, CCR2, AGTR2, and CXCL1 were significantly enriched in the inflammatory response, cytokine–cytokine receptor interaction, chemokine signaling, and neuro-active ligand–receptor interaction pathways. Furthermore, this study identified essential miRNAs such as miR-3123, miR-548g-3p, and miR-9-5p [55]. Yang et al. (2022) pointed out four other hub genes, including CXCL12, LTBP1, LOXL1, and IGFBP3 [56]. The cytokine CXCL12, which recruits monocytes and lymphocytes, was also indicated by Liu et al. (2021) as a hub DEG in AF along with IL7R, TNFSF13B, and CD8A associated with regulatory T cells or natural killer-cell-activated immune cells [57]. These data show that CXCL12 plays a vital role in regulating the local inflammatory response and immune cell infiltration. Thus, this cytokine may be used as a biomarker to distinguish subsets of AF. Interestingly, Lal et al. (2022) indicated metformin as a top repurposed drug candidate for AF, discussing the proximity of NPPB (brain natriuretic peptide) and CXCL12 to the metformin target DPP4. Brain natriuretic peptide suppresses the renin–aldosterone–angiotensin system activity, reducing the activation of nuclear factor-kB and the production of interleukin-6 and C-reactive protein, and consequently, oxidative stress [58].

## 6. Integrating Data Using Bioinformatic Tools

Data integration allowed the recognition of 1735 biological entities such as lncRNAs, miRNAs, genes, and proteins discussed among all 28 references. Table 1 presents the top 30 biological entities cited in at least five articles and their related information. Figure 2 shows the respective protein–protein network that may allow the determination of topological parameters useful for disease modeling.

Table 2 presents data from enrichment analysis concerning ontology pathways. The top 30 biological entities mined from AF network studies are mainly related to the inflammatory process and the immune system.

## 7. Conclusions and Perspectives

This review presents the outstanding contribution of network medicine in providing interesting insights into AF. It highlights the substantial role of structural remodeling, the immune system, and inflammation in this disease etiology. In addition, this work demonstrates how bioinformatic tools can be used in network medicine. Data integration suggests the protagonism of the immune system and inflammation over structural remodeling.

Holistic knowledge of the underlying mechanisms of AF is crucial for treatment improvement as the genes identified may serve as drug targets for new or repurposed drugs. The relevant participation of inflammatory processes in AF physiopathology allows discussion of the potential of anti-inflammatory treatment, for example. However, none of the papers in this review studied AF considering gender or ethnic differences. Thus, there are still gaps to be filled regarding the disease.

## Figures and Tables

**Figure 1 life-13-01364-f001:**
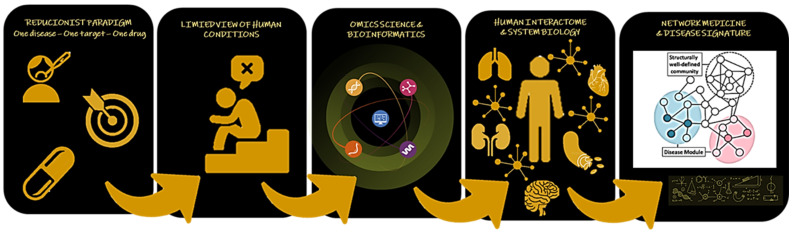
Scheme illustrating the contribution of omics sciences for a holistic point of view about the health–disease process. Based on the interactome, molecular interactions can be mapped and visualized through networks. The nodes represent individual biological entities, and the edges mean connections. Thus, network medicine interprets disease traits as perturbations within the interactome. With this approach, potential disease drivers are unveiled, and the effect of drugs, novel or repurposed, used alone or in combination, may be studied, allowing advances in the knowledge of disease mechanisms and precision therapeutics.

**Figure 2 life-13-01364-f002:**
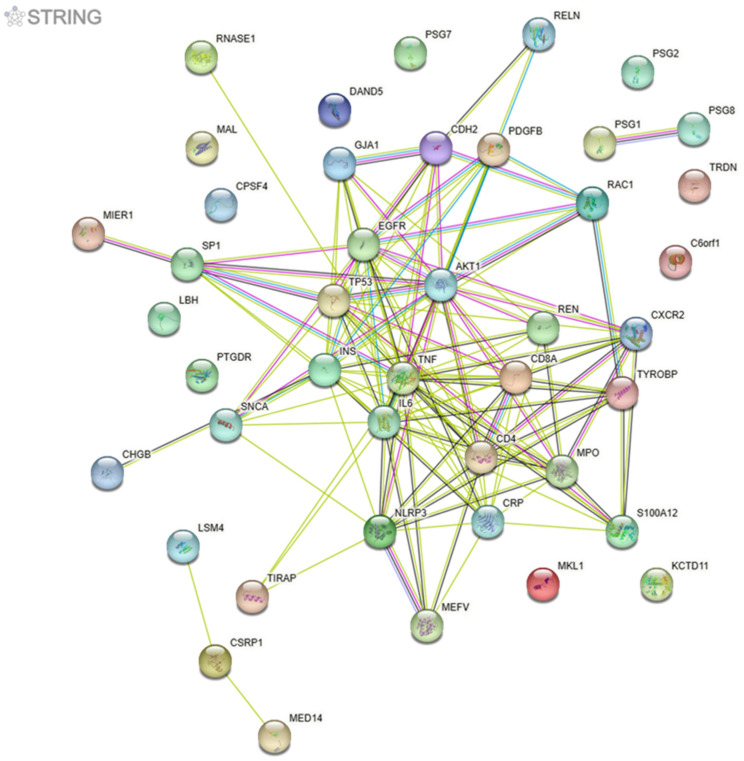
Network encompassing the top 30 biological entities shown in Table 1. Network nodes (circles) represent all the proteins produced by a single protein-coding gene locus. Edges represent protein–protein-specific and meaningful associations, i.e., functional and physical interactions, with an evidence score of 0.900. Colored nodes represent query proteins and the first shell of interactors, while white nodes encompass the second shell of interactors. Empty nodes refer to proteins with an unknown 3D structure; filled nodes have some 3D structure known or predicted.

**Table 1 life-13-01364-t001:** Top 30 biological entities revealed by literature mining.

from	Entry	Protein Names	Gene Names
ENSP00000292476	O95639	Cleavage and polyadenylation specificity factor subunit 4	CPSF4 CPSF30 NAR NEB1
ENSP00000398698	P01375	Tumor necrosis factor	TNF TNFA TNFSF2
ENSP00000255030	P02741	C-reactive protein	CRP PTX1
ENSP00000385675	P05231	Interleukin-6	IL6 IFNB2
ENSP00000011653	P01730	T-cell surface glycoprotein CD4	CD4
ENSP00000272190	P00797	Renin	REN
ENSP00000330382	P01127	Platelet-derived growth factor subunit B	PDGFB PDGF2 SIS
ENSP00000356275	P21291	Cysteine and glycine-rich protein 1	CSRP1 CSRP CYRP
ENSP00000386559	P01732	T-cell surface glycoprotein CD8 alpha chain	CD8A MAL
ENSP00000451828	P31749	RAC-alpha serine/threonine-protein kinase	AKT1 PKB RAC
ENSP00000282561	P17302	Gap junction alpha-1 protein	GJA1 GJAL
ENSP00000380432	P01308	Insulin	INS
ENSP00000269141	P19022	Cadherin-2	CDH2 CDHN NCAD
ENSP00000275493	P00533	Epidermal growth factor receptor	EGFR ERBB ERBB1 HER1
ENSP00000319635	P25025	C-X-C chemokine receptor type 2	CXCR2 IL8RB
ENSP00000348461	P63000	Ras-related C3 botulinum toxin substrate 1	RAC1 TC25 MIG5
ENSP00000378733	Q53QV2	Protein LBH	LBH
ENSP00000381057	P07998	Ribonuclease pancreatic	RNASE1 RIB1 RNS1
ENSP00000417604	Q86T20	Small integral membrane protein 29	SMIM29 C6orf1 LBH
ENSP00000225275	P05164	Myeloperoxidase	MPO
ENSP00000262629	O43914	TYRO protein tyrosine kinase-binding protein	TYROBP DAP12 KARAP
ENSP00000269305	P04637	Cellular tumor antigen p53	TP53 P53
ENSP00000303424	Q13258	Prostaglandin D2 receptor	PTGDR
ENSP00000308970	P11464	Pregnancy-specific beta-1-glycoprotein 1	PSG1 B1G1 PSBG1 PSGGA
ENSP00000323155	Q8N907	DAN domain family member 5	DAND5 CER2 CKTSF1B3 GREM3 SP1
ENSP00000329357	P08047	Transcription factor Sp1	SP1 TSFP1
ENSP00000337383	A0A7I2R3P8	NLR family pyrin domain containing 3	NLRP3
ENSP00000338345	P37840	Alpha-synuclein	SNCA NACP PARK1
ENSP00000357726	P80511	Protein S100-A12	S100A12
ENSP00000368244	P05060	Secretogranin-1	CHGB SCG1

**Table 2 life-13-01364-t002:** Data from enrichment analysis of the top 30 biological entities revealed by literature mining.

Term ID	Term Name	*p*-Value	−log10 (*p*-Value)
GO:0044419	biological process involved in interspecies interaction between organisms	2.85 × 10^−7^	6.5
GO:0006954	inflammatory response	4.07 × 10^−7^	6.4
GO:0001775	cell activation	4.38 × 10^−7^	6.4
GO:0048584	positive regulation of response to stimulus	4.80 × 10^−7^	6.3
GO:0009605	response to external stimulus	6.73 × 10^−7^	6.2
GO:2000377	regulation of reactive oxygen species metabolic process	6.92 × 10^−7^	6.2
GO:0032501	multicellular organismal process	9.69 × 10^−7^	6
GO:0072593	reactive oxygen species metabolic process	1.03 × 10^−6^	6
GO:0002376	immune system process	1.47 × 10^−6^	5.8
GO:0051239	regulation of multicellular organismal process	1.80 × 10^−6^	5.7

## Data Availability

No new data were created or analyzed in this study. Data sharing is not applicable to this article.

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
