# Peer review of "Reviewing Atrial Fibrillation Pathophysiology from a Network Medicine Perspective: The Relevance of Structural Remodeling, Inflammation, and the Immune System"

_life, 2023, doi:10.3390/life13061364_

Round 1

Reviewer 1 Report

Life 2350252 peer review

This article tries to review atrial fibrillation pathophysiology under a new perspective based on “network medicine”. The term "network medicine" is here used a synonym of "systems biology" (erroneously called here "system biology") i.e. the study of complex systems by a holistic approach which makes large use of omics. Undoubtedly, it is a rather ambitious task, which requires the ability of carrying out a high-level discussion contrasting most commonly accepted viewpoints. Such aim is hampered by the imperfect use of the English language, which makes it difficult to understand even the description of the state of art, as in the Introduction. For instance, the sentence “…in lowering the risk of safety and efficiency endpoint events.” (Page 2 line 62) is contradictory within itself, since safety and efficiency are not risks. Because of this, the Reader wonders what the Authors really mean and is distracted from following the discussion. I therefore strongly recommend enlisting the help of a native English speaker expert able to convey the scientific message.

There is no “Methods” section. I suggest establishing it and grouping there the information about the methods whereby the publications were selected (now in section 3) and the way the information contained therein was integrated by bioinformatics tools (now in section 6). Moreover, more detail should be given about the way such bioinformatics tools were used. While Authors used Gene Ontology, which is a valid tool for identifying gene functions, KEGG analysis, which suggests involved pathways, should be used as well.

Results are presented in a disorganized way. I suggest grouping them according to the main pathways identified. Sections 4 and 5 deal with structural remodeling and inflammation and immune system involvement, respectively. In an analogous way, there should be paragraphs about the several pathways described in sections 2 and 3. Moreover, section 4 describes pathways unrelated with structural remodeling, whilst structural remodeling is described also in section 3.  The conclusion that structural remodeling, inflammation and immune system are the most relevant pathways should be reached as a result of the bioinformatics analysis rather than as a result of critical literature reading, as it appears.

Figures 2 and 3 show “biological entities” as small colored balls without any explanation about the color code. The low resolution of the figure does not allow reading the symbols within the balls, therefore the Figures convey very little information. I suggest coloring the balls according to the identified pathway the “entities” belong with; if the intent of the Authors was different, they should nevertheless add figures showing these pathways. Non-coding RNAs are surely part of these networks, as discussed in section 3, but they do not appear in Figures 2 and 3, as far as one can understand.

It is possible that some of the criticisms I raised are due to the poor ability to represent the work undertaken, rather than being intrinsic, as I pointed out above.

The imperfect use of the English language makes it difficult to understand even the description of the state of art, as in the Introduction. For instance, the sentence “…in lowering the risk of safety and efficiency endpoint events.” (Page 2 line 62) is contradictory within itself, since safety and efficiency are not risks. Because of this, the Reader wonders what the Authors really mean and is distracted from following the discussion. I therefore strongly recommend enlisting the help of a native English speaker expert able to convey the scientific message.

Author Response

  • Comment 1: This article tries to review atrial fibrillation pathophysiology under a new perspective based on “network medicine”. The term "network medicine" is here used a synonym of "systems biology" (erroneously called here "system biology") i.e. the study of complex systems by a holistic approach which makes large use of omics. Undoubtedly, it is a rather ambitious task, which requires the ability of carrying out a high-level discussion contrasting most commonly accepted viewpoints. Such aim is hampered by the imperfect use of the English language, which makes it difficult to understand even the description of the state of art, as in the Introduction. For instance, the sentence “…in lowering the risk of safety and efficiency endpoint events.” (Page 2 line 62) is contradictory within itself, since safety and efficiency are not risks. Because of this, the Reader wonders what the Authors really mean and is distracted from following the discussion. I therefore strongly recommend enlisting the help of a native English speaker expert able to convey the scientific message.

Response: Thanks for pointing this out. We agree with this comment. Indeed, "systems biology" is the correct term encompassing the study of complex systems by a holistic approach using omics sciences. Therefore, we have adjusted the term within the document. In addition, we rewrote the text aiming to improve the English language by gaining clarity. The paper has undergone English language editing by MDPI. We have also tried to present a more suitable differentiation among "systems biology," "network biology," "network medicine," and "network pharmacology" concepts. The network biology approach includes network construction and topological analysis of data provided by omics sciences encompassing systems biology concepts. Not necessarily focusing on disease traits and their treatment, the perspective of network medicine, but on the interactome. The alternative term network pharmacology embraces studies focused on pharmacological treatment. However, regarding the sentence quoted in the commentary, it was stated just like the quoted reference. Nevertheless, understanding the reviewer's point, we changed the related text.

  • Comment 2: There is no “Methods” section. I suggest establishing it and grouping there the information about the methods whereby the publications were selected (now in section 3) and the way the information contained therein was integrated by bioinformatics tools (now in section 6). Moreover, more detail should be given about the way such bioinformatics tools were used. While Authors used Gene Ontology, which is a valid tool for identifying gene functions, KEGG analysis, which suggests involved pathways, should be used as well.

Response: Agree. We have, accordingly, included the section "Materials and Methods." However, KEGG analysis is not crucial to answer the secondary goal of this review. According to Fran Supek and Nives Škunca (Visualizing GO Annotations. Methods Mol Biol. 2017;1446:207-220. doi: 10.1007/978-1-4939-3743-1_15), Gene Ontology (GO) terms typically describe functions of genes datasets subjected to statistical tests for enrichment for various functional categories. Gene datasets can indicate that a systems-level change in a particular condition has occurred, which may be described by examining the common properties of the differential expressed genes. For example, the GO enrichment analysis may answer if the genes participate in the same metabolic or signaling pathway and perform similar biochemical functions. They present KEGG Pathways as an alternative. This way, we adjust the text to clearly state that the GO enrichment analysis presented refers to ontology pathways. We have substituted the reference quoted previously with the reference above. We think it is pertinent to state in the last paragraph of the section before "Materials and Methods" that this review aims to present network medicine as an interesting approach to helping researchers understand AF more deeply, describe the main related findings in the literature, and demonstrate briefly how bioinformatic tools can provide a holistic perspective of the disease's underlying mechanisms, favoring treatment improvement.

  • Comment 3: Results are presented in a disorganized way. I suggest grouping them according to the main pathways identified. Sections 4 and 5 deal with structural remodeling and inflammation and immune system involvement, respectively. In an analogous way, there should be paragraphs about the several pathways described in sections 2 and 3. Moreover, section 4 describes pathways unrelated with structural remodeling, whilst structural remodeling is described also in section 3. The conclusion that structural remodeling, inflammation and immune system are the most relevant pathways should be reached as a result of the bioinformatics analysis rather than as a result of critical literature reading, as it appears.

Response: Thanks for this suggestion. It would have been interesting to explore this aspect. However, in the case of our study, it seems slightly out of scope because the aim is to present network medicine as an exciting approach to helping researchers to understand AF more deeply, describing previously the main related findings disposed of in the literature. The papers consulted have used bioinformatics analysis to provide this information; thus, the conclusion that structural remodeling, inflammation, and immune system are the most relevant pathways is because of critical literature reading. However, this review also aims to demonstrate bioinformatic tools favoring network medicine through data integration of the presented literature. Because of this, we did not rearrange the results section. Data integration suggests the protagonism of the immune system and inflammation over structural remodeling.

  • Comment 4: Figures 2 and 3 show “biological entities” as small colored balls without any explanation about the color code. The low resolution of the figure does not allow reading the symbols within the balls, therefore the Figures convey very little information. I suggest coloring the balls according to the identified pathway the “entities” belong with; if the intent of the Authors was different, they should nevertheless add figures showing these pathways. Non-coding RNAs are surely part of these networks, as discussed in section 3, but they do not appear in Figures 2 and 3, as far as one can understand.

Response: We agree with this and have incorporated the suggestion throughout the manuscript. However, it is generally difficult to identify individual nodes in dense networks shown throughout network medicine papers. We invite the reviewer to look for an example presented in Figure 3 from the paper of Castillo-Velázquez R et al. published in 2023 that used the STRING app contained in OnTheFly2.0 (Bioinformatic prediction of the molecular links between Alzheimer’s disease and diabetes mellitus. PeerJ 11:e14738 https://doi.org/10.7717/peerj.14738). Considering the low contribution of Figure 2 for the paper, we decided to remove it. We have maintained Figure 3, attaching it with a more informative legend. It is essential to highlight that non-coding RNAs are not part of the network presented because any specific non-coding RNA was within the top 30 biological entities described in at least five papers consulted and presented in Table 1.

Reviewer 2 Report

The presented work aims to look at the pathology of AF from the perspective of network medicine, in order to help researchers to understand the disease more deeply.

The paper is very extensive, detailed, technically adequately conceived, but thematically interesting only to experts/scientists who deal with/are interested in network medicine.

Personally, I read only 2 chapters with interest and understanding: 1) introduction and 2) conclusion.

.

Author Response

Reviewer 2 also captured the aim of the original proposal. He/she commented about the goal of looking at the pathology of AF from the perspective of network medicine to help researchers to understand the disease more deeply. According to him/her, the paper is detailed and technically adequately conceived. However, unlike Reviewer 3, he/she thinks the paper would be thematically interesting only to experts/scientists who deal with/are interested in network medicine. It is essential to mention that, according to Ms. Viola Wu, Section Managing Editor, the topic fits the Special Issue titled "Mechanisms of Atrial Fibrillation." We had previously briefed her about the proposal as we were attending an invitation to submit a review article without a fee. It is our understanding, and we think that is also hers, that this perspective may be interesting for most AF researchers. This Research Topic welcomes basic, translational, clinical, and applied research that improves understanding of AF. Intracellular signaling and regulation of gene transcription constitute a potential area of interest, but the Special Issue should not be limited to it. Reviewer 2 did not make any additional comments that would guide us to adjust the paper beyond English. The paper has undergone English language editing by MDPI. 

Reviewer 3 Report

The article approaches atrial fibrillation from another point of view, which makes it extremely interesting and actual offering new future perspectives regarding the therapeutic approach of atrial fibrillation.

The review presents the contribution of network medicine in providing interesting insights about atrial fibrillation, highlighting the role of structural remodeling, immune system, and inflammation in its etiology. Another important point of this review  demonstrates how bioinformatic tools can be used in network medicine. Holistic approach of the underlying mechanisms of atrial fibrillation is crucial for treatment improvement as the genes identified may serve as drug targets for new therapies.

This work aims to review the pathology of atrial fibrillation from a network medicine perspective, helping the researchers to study the disease more deeply. Essential concepts involved in network medicine are highlighted. All data together show the substantial role of structural  remodeling, immune system, and inflammation in this disease etiology.

But, references are not uniformly written, and references no 11,13,14, and 15 are very old and not absolutely necessary for the text. Try to replace them with new data.

English language may be improved

Author Response

According to Reviewer 3, the article's approach is interesting and offers new perspectives regarding the therapeutic approach to atrial fibrillation (AF). He/she has highlighted the relevance of demonstrating how bioinformatic tools can be used in network medicine to provide a holistic perspective of the underlying mechanisms of AF for treatment improvement. Reviewer 3 captured the goal of the original proposal, which aims to review the pathology of AF from a network medicine perspective, helping the researchers to study the disease more deeply. According to his/her perception, essential concepts involved in network medicine are presented so that all data show the substantial role of structural remodeling, immune system, and inflammation in this disease etiology. Reviewer 3 also pointed out that references need to be uniformly written, and references no, 11,13,14, and 15 are very old and unnecessary for the text. Therefore, we adjusted them as suggested. The paper has also undergone English language editing by MDPI. 

Round 2

Reviewer 1 Report

This reviewer believes that the manuscript has been substantially ameliorated and that all concerns raised in the previous comment have been addressed and solved in a satisfactory way